# Neutrophil Extracellular Traps in the Pathogenesis of Equine Recurrent Uveitis (ERU)

**DOI:** 10.3390/cells8121528

**Published:** 2019-11-27

**Authors:** Leonie Fingerhut, Bernhard Ohnesorge, Myriam von Borstel, Ariane Schumski, Katrin Strutzberg-Minder, Matthias Mörgelin, Cornelia A. Deeg, Henk P. Haagsman, Andreas Beineke, Maren von Köckritz-Blickwede, Nicole de Buhr

**Affiliations:** 1Department of Physiological Chemistry, University of Veterinary Medicine Hannover, Bünteweg 17, D-30559 Hannover, Germany; leonie.fingerhut@tiho-hannover.de (L.F.); Ariane.Schumski@med.uni-muenchen.de (A.S.); 2Research Center for Emerging Infections and Zoonoses (RIZ), University of Veterinary Medicine Hannover, Bünteweg 17, D-30559 Hannover, Germany; 3Clinic for Horses, University of Veterinary Medicine Hannover, Bünteweg 9, D-30559 Hannover, Germany; bernhard.ohnesorge@tiho-hannover.de; 4Tierärztliche Gemeinschaftspraxis für Pferde Wedemark, Lange Loh 15, D-30900 Wedemark, Germany; info@pferdepraxis-wedemark.de; 5IVD Gesellschaft für Innovative Veterinärdiagnostik mbH (IVD GmbH), Albert-Einstein-Str. 5, D-30926 Seelze, Germany; strutzberg@ivd-gmbh.de; 6Colzyx AB, Medicon Village, SE-223 81 Lund, Sweden; matthias@colzyx.com; 7Chair of Animal Physiology, Department of Veterinary Sciences, LMU Munich, Lena Christ Str. 48, D-82152 Martinsried, Germany; Cornelia.Deeg@lmu.de; 8Department of Infectious Diseases & Immunology, Faculty of Veterinary Medicine, Utrecht University, Yalelaan 1, 3584 CL Utrecht, The Netherlands; H.P.Haagsman@uu.nl; 9Department of Pathology, University of Veterinary Medicine Hannover, Bünteweg 17, D-30559 Hannover, Germany; andreas.beineke@tiho-hannover.de

**Keywords:** NETs, equine recurrent uveitis, horse, cathelicidin

## Abstract

Equine recurrent uveitis (ERU) is considered one of the most important eye diseases in horses and typically appears with relapsing inflammatory episodes without systemic effects. Various disorders have been described as an initial trigger, including infections. Independent of the initiating cause, there are numerous indications that ERU is an immune-mediated disease. We investigated whether neutrophil extracellular traps (NETs) are part of the ERU pathogenesis. Therefore, vitreous body fluids (VBF), sera, and histological sections of the eye from ERU-diseased horses were analyzed for the presence of NET markers and compared with horses with healthy eyes. In addition, NET formation by blood derived neutrophils was investigated in the presence of VBF derived from horses with healthy eyes versus ERU-diseased horses using immunofluorescence microscopy. Interestingly, NET markers like free DNA, histone-complexes, and myeloperoxidase were detected in higher amounts in samples from ERU-diseased horses. Furthermore, in vitro NET formation was higher in neutrophils incubated with VBF from diseased horses compared with those animals with healthy eyes. Finally, we characterized the ability of equine cathelicidins to induce NETs, as potential NET inducing factors in ERU-diseased horses. In summary, our findings lead to the hypothesis that ERU-diseased horses develop more NETs and that these may contribute to the pathogenesis of ERU.

## 1. Introduction

Equine recurrent uveitis (ERU) is considered one of the most important eye diseases in horses, affecting between 2 and 25 percent of animals [1,2,3,4]. Common clinical findings include corneal haze, aqueous flare, pupil miosis, and opacified vitreous body fluid (VBF) with deposits. This serofibrinous and lymphoplasmacellular inflammation often results in chronic alterations such as synechiae, cataracts, lens luxation, retinal detachment, and phthisis bulbi [5,6,7]. Eventually, the cumulative degeneration of ocular structures can lead to blindness. Thus, a diagnosis can be made when a horse is presented with relapsing uveitic findings without systemic effects. This disease can be divided into three different types, the most typical in Europe being the posterior form affecting predominantly vitreous body, retina, and choroid [7]. The initial cause of the uveitis remains obscure, but genetic and autoimmune components were found [8,9,10]. In a genome-wide association study, a single nucleotide polymorphism on chromosome 20 was discovered to be associated with ERU [11]. Additionally, *Leptospira* spp. can be detected in about 60 percent of the patients [12,13,14,15]. Whether those pathogens cause the destruction of the blood-retina barrier or the barrier is destroyed first, thus enabling pathogens to enter the immune-privileged organ, is under discussion [16]. The treatment options range from immunosuppressive medication to different surgical procedures, for instance, vitrectomy. Hereby, vitreous body fluid is exchanged in a minimally invasive way by buffered salt solution with or without antibiotics.

As autoimmune processes are discussed as being part of ERU, it is of interest that a defense mechanism of neutrophils, neutrophil extracellular traps (NETs) formation, is described as being involved in autoimmune diseases [17,18,19]. Besides phagocytosis and degranulation, NET formation is another strategy of neutrophils against invading pathogens, also referred to as NETosis [20]. NET formation was explained mainly by two different mechanisms [21,22,23,24]. The ‘suicidal’ NETosis is a synonym for the lytic NET release, leading to dead neutrophils after several hours. The ‘vital’ NETosis is characterized by the rapid release of NETs, and neutrophils undergoing this mechanism are still able to phagocyte or degranulate [25]. NET release by viable cells is mediated by a vesicular mechanism and reactive oxygen independent [23]. Furthermore, NET release by viable cells in the form of mitochondrial DNA has been described. NETs, independent of the mechanism, consist of decondensed chromatin, histones, antimicrobial peptides (AMPs), and granule proteins [21,22]. The contained AMPs play an important role in the formation and antimicrobial function of NETs. These components build web-like structures to entrap and kill microbes [21]. Host nucleases are crucial for maintaining the balance between NET formation and elimination, and hence for preventing accumulation of NETs [26]. On the other hand, a detrimental role of NETs has been detected in non-infectious conditions, such as autoimmune or chronic diseases, thrombosis, and cancer. For instance, NETs contribute to the pathogenesis of systemic lupus erythematosus, psoriasis, or rheumatoid arthritis by autoantigen exposition [18,27,28]. Moreover, the involvement of NETs and associated proteins in bacterial keratitis owing to *Pseudomonas aeruginosa* ocular biofilms and in the ocular graft-versus-host disease dry eye in humans, with both diseases affecting the ocular surface, has been proven [29,30]. Barliya et al. [31] demonstrated intraocular NET induction through cytokines, namely interleukin-8 (IL-8) and tumor necrosis factor α (TNF-α), in a murine model. Furthermore, they showed the occurrence of NETs in human vitreous body fluid and other ocular components in proliferative diabetic retinopathy, to a higher extent in more severe cases [31]. The existence of NETs in VBF of such patients, as well as in diabetic rats, has recently been confirmed by Wang et al. [32]. Whether NETs contribute to the pathogenesis of ERU has not yet been investigated.

Thus, it seems obvious to assume a potential role of NETs or the associated AMPs in the pathogenesis of ERU. Interestingly, the closest genes to the single nucleotide polymorphism found to be linked with ERU are IL-17A and IL-17F [11]. This proinflammatory family of cytokines was recently reported to modulate NET formation and AMP production [27,33]. Furthermore, IL-17 occurs with an elevated tissue expression in human autoimmune uveitis [34]. Chen et al. [35] suggest an inductive effect of IL-17 on the expression of the human cathelicidin LL-37. Cathelicidins are a subtype of AMPs and three different sequences can be found in equine bone marrow RNA, referred to as eCATH 1-3. However, only two pro-peptides are cleaved inside neutrophils into the mature peptides eCATH 2 and 3 [36].

The aim of this study was to first clarify the appearance of NETs during ERU, as well as the involvement of associated AMPs in the pathogenesis of this commonly occurring disease. The focus within the AMPs was on the equine cathelicidins owing to their possible connection to IL-17 and the genetic components of ERU.

## 2. Materials and Methods

### 2.1. Samples

In the conducted experiments, samples from two different clinics were analyzed. In study part I, serum samples obtained in Munich, Germany, were investigated in quantitative measurements of free DNA and nuclease activity, comparing healthy eyes of horses with those of ERU-diseased horses (Figure 1). Furthermore, these serum samples were used in an ELISA for equine cathelicidins (Figure A2a). All other experiments (study part II) were performed using vitreous body fluid (VBF) and blood samples for neutrophil isolation from Hannover, Germany.

#### 2.1.1. Study Part I (Munich)

The serum samples included in the analysis of free DNA and nuclease activity were obtained from horses vitrectomized owing to diagnosis of ERU at the Equine Hospital of the Faculty of Veterinary Medicine, LMU Munich in Munich, Germany, until 2012. The control samples were taken from those animals with healthy eyes. The study protocol for obtaining blood samples from ERU cases in the quiescent stage of the disease and controls (both obtained from the Equine Clinic, LMU Munich) was permitted by the Ethical Committee of Upper Bavaria’s Regional Government (Regierung von Oberbayern; permit number: ROB-55.2Vet-2532.Vet_03-17-88). All experiments were performed in accordance with the relevant guidelines and regulations. The samples were stored at −20 °C until further analysis.

#### 2.1.2. Study Part II (Hannover)

##### Blood Samples for Neutrophil Isolation

Blood collection from healthy horses was approved by Lower Saxony State Office for Consumer Protection and Food Safety (LAVES) (Niedersächsisches Landesamt für Verbraucherschutz und Lebensmittelsicherheit) under nos. 12A243 and 18A302 for study part II. Fresh equine neutrophils were isolated from these lithium-heparinized blood samples.

##### VBF Samples Obtained from Vitrectomy Patients and Healthy Horses

The VBF samples of ERU-diseased horses included in the NET induction assay, ELISA to detect equine cathelicidins, and electron microscopy evaluation were obtained from horses vitrectomized between 2016 and 2018 at the Clinic for Horses of the University of Veterinary Medicine Hannover, Germany. The pre-examinations, treatment, and surgical procedure were performed as previously published by von Borstel et al. and Baake et al. [6,37]. Remnant VBF samples from diagnostic procedures were used in this study. The control samples were derived post mortem from horses without ophthalmologic findings indicating ERU, which had been euthanized because of other reasons between 2016 and 2017. The euthanasia of these horses had been approved and registered by the local Animal Welfare Officer in accordance with the German Animal Welfare Law under number TiHo-T-2016-4. The eyes from these horses were enucleated and freed from the surrounding tissues, and VBF was extracted with a 5 mL syringe through a cross-opening cut into the eye. These patients are described in more detail in Table A1. Signalment and clinical data were obtained from the medical records.

The samples of ERU-diseased horses were chosen randomly from a collection of samples taken between 2016 and 2019. Inclusion criteria were a positive microscopic agglutination test (MAT) for intraocular leptospiral antibodies and a positive result of a polymerase chain reaction (PCR) detecting leptospiral DNA. This analysis is routinely conducted in VBF that is completely exchanged and collected in all ERU-diseased horses during vitrectomy. The VBF of included control horses was analyzed in a similar manner and VBF was included if negative MAT and PCR results were found. The samples were stored at −80 °C until further analysis. MAT and PCR were conducted at IVD GmbH, Seelze, Germany.

##### Samples Obtained from Enucleations

The paraffin sections and vitreous body fluid for NET detection ex vivo and in situ were received from eyes that had been enucleated at the Clinic for Horses of the University of Veterinary Medicine Hannover, Germany, in 2019 as part of curative surgery. A description of signalment, pathological findings, and the results of the screening for *Leptospira* by MAT and PCR are displayed in Table A2. Signalment and clinical data were obtained from the medical records. The eyes were freed from the surrounding tissues and 1-5 mL VBF was extracted with a 5 mL syringe through a cross-opening cut into the eye. VBF was immediately put on slides (8 mm, Thermo Fisher, Waltham, MA, USA CB00080RA120) coated beforehand with 0.01% poly-l-lysine (Sigma-Aldrich, St. Louis, MO, USA P4707) for 20 min. They were then centrifuged for five minutes at 370 g at room temperature and afterwards fixated with paraformaldehyde (Science Services E15710-250) to a final concentration of 4%. To fixate the eye, it was flushed first with phosphate-buffered formalin (10%, 3 × 5 mL) by injection through a needle (21G, WDT 07391) at the opposite side of the first insertion and then put in formalin and fixed for two to four days. Samples were dehydrated in a graded series of ethanol and embedded via xylene in paraffin wax and cut into 4 µm sections with a microtome. Slides were deparaffinised and hydrated through descending concentrations of ethanol. Afterwards, sections were stained with hematoxylin-eosin (HE, hemalaun according to Delafield) following a routine protocol. Histology slices were examined with a standard light microscope.

### 2.2. Pico Green Assay

The amount of free DNA in serum of horses with healthy eyes and ERU-diseased horses was evaluated by a Pico Green assay using a plate reader (FLUOStar, Optima) in study part I. Therefore, 1:200 in Tris-EDTA buffer solution (TE) diluted PicoGreen (Quant-iT PicoGreen Invitrogen, Carlsbad, CA, USA P11496) was mixed 1:2 with serum in a 96 black flat bottom well plate (BRANDplates^®^ 781608). A dilution series of calf thymus DNA (Sigma-Aldrich D3664, 1 mg/mL) was used for a standard curve. The plate was measured after a five minute incubation period at room temperature in the dark.

### 2.3. Nuclease Activity Assay

#### 2.3.1. Quantitative Measurement

The nuclease activity in serum was examined in study part I by mixing 7.5 µg of calf thymus DNA (Sigma-Aldrich D3664, 1 mg/mL) with 10 µL serum and 40 µL Tris. Respective controls were included in each run. As negative control of 7.5 µg of calf thymus DNA was mixed with 50 µL Tris, as positive control of 7.5 µg of calf thymus DNA was mixed with 40 µL Tris and 10 µL DNaseI (2 U/mL). Samples were incubated for six hours at 37 °C. Afterwards, intact DNA was precipitated with phenol-chloroform. The DNA was visualized on 1% agarose gel containing Roti^®^-GelStain (Carl Roth 3865.1) by gel electrophoresis. The relative nuclease activity was determined by comparison with a standard row of nuclease activity (DNaseI 0, 0.015, 0.03, 0.06, 0.125, 0.25, 0.5, and 1 U/mL, respectively).

#### 2.3.2. Qualitative Measurement

The nuclease activity in serum and VBF of healthy controls and ERU patients were examined in study part II by mixing 0.5 µg of calf thymus DNA (Sigma-Aldrich D3664, 1 mg/mL) with 50 µL of the sample (serum or VBF). This was then incubated at 37 °C for 20 h. Afterwards, 1% agarose gel electrophoresis was conducted with DNA staining with Roti^®^-GelStain (Carl Roth 3865.1). A 1 kb marker (Invitrogen 1 kb Plus DNA Ladder 10787-018, 1 µg/µL), as well as a negative and positive control with Hank’s Balanced Salt Solution (GIBCO^®^HBSS 14025050) and 0.5 µg calf thymus DNA, with or without 0.5 µL micrococcal nuclease (Sigma-Aldrich N5386, 1 Units/µL), were added.

### 2.4. PMN Isolation

In study part II, fresh equine neutrophils were isolated from lithium-heparinized blood using Biocoll density gradient and lysis of erythrocytes. All steps were performed in duplicate. Thereby, the blood was diluted 2.5:1 in phosphate-buffered saline (Sigma-Aldrich P5493-1L, 10× PBS diluted in endotoxin-free water to 1× concentrated), layered onto the Biocoll (Biochrom AG L6115, 1.077 g/mL), and then centrifuged for 20 min at 400 *g* at room temperature without using a centrifuge brake. Afterwards, the supernatant was removed and cold endotoxin-free 0.2% NaCl-solution was added to the sediment for 30 s and carefully inverted. The lysis of the erythrocytes was then stopped by adding cold endotoxin-free 1.6% NaCl-solution. This was then centrifuged at 250 *g* for six minutes at 4 °C. The lysis step was repeated once or twice depending on the clarity of the obtained pellet. The two pellets were finally pooled and resuspended in cold Roswell Park Memorial Institute medium (RPMI, gibco 11835063).

### 2.5. NET Induction

Within study part II, fresh isolated blood-derived equine neutrophils were seeded in a concentration of 2 × 10^6^ cells mL^−1^ (2 × 10^5^ cells per well) on slides (8 mm, Thermo Fisher CB00080RA120), which had previously been coated for 20 min with 0.01% poly-l-lysine (Sigma-Aldrich P4707). Then, 100 µL stimulus was added to each well and incubated for 240 min at 37 °C with 5% CO_2_. The well plate was then centrifuged for five minutes with 370 g at room temperature and, afterwards, the samples were fixated with paraformaldehyde (Science Services E15710-250, final concentration of 3.2%) and stored at 4 °C. The examined stimuli included the equine cathelicidins (eCATH 1, eCATH 2, eCATH 3 [38]), each in a 5 µM and 10 µM concentration, in different environments. The first set-up analyzed the equine cathelicidins in the presence of RPMI (gibco 11835063), using RPMI alone as a negative control and methyl-β-cyclodextrin (Sigma-Aldrich C4555-1G, final 10 mM) diluted in RPMI as a positive control. The next five independent experiments were conducted in the presence of VBF from the control horses with healthy eyes, with an additional control of VBF alone. The methyl-β-cyclodextrin was hereby diluted in VBF. A set-up of seven independent experiments using VBF obtained from ERU patients was also performed.

### 2.6. Immunofluorescence Staining of NETs

The samples in study part II were permeabilized for five minutes with Triton X-100 (Sigma-Aldrich T8787-50ML) for the co-staining of DNA/histone-1 and myeloperoxidase as NET markers. Then, they were blocked for 20 min with 3% normal donkey serum (Sigma-Aldrich D9663-10ML), 3% cold water fish gelatine (Sigma-Aldrich G7041-100G), 1% bovine serum albumin (Albumin fraction V, Roth 2923225), and 0.05% Tween20 (Roth 9127.2) in 1× phosphate-buffered saline (PBS, Sigma-Aldrich P5493-1l, 10× PBS diluted to 1× PBS in distilled water). Afterwards, the samples were incubated for one hour with a monoclonal mouse anti-DNA/histone-1 antibody (Millipore, Burlington, MA, USA MAB3864, 0.55 mg/mL) diluted 1:1000 in blocking buffer and a rabbit anti-human myeloperoxidase antibody (Dako A0398, 3.2 mg/mL) diluted 1:300 in blocking buffer. Isotype controls were incubated with an IgG2α antibody from murine myeloma (Sigma-Aldrich M5409-.1MG, 0.2 mg/mL) diluted 1:364 in blocking buffer and IgG from rabbit serum (Sigma-Aldrich I5006, 10.4 mg/mL) diluted 1:975 in blocking buffer. After washing, a goat anti-mouse DyLight 488 antibody (Thermo Scientific 35503, 1 mg/mL) diluted 1:1000 and a goat anti-rabbit Alexa 633 antibody (Thermo Scientific A 21070, 2 mg/mL) diluted 1:500, both in blocking buffer, were used to perform the secondary staining for one hour in the dark. After washing, staining with aqueous Hoechst 33342 (Thermo Fisher 62249, 1:1000 in aqua dist.) was carried out for ten minutes. The slides were washed and embedded in ProLong^®^Gold antifade reagent (Invitrogen P36930).

### 2.7. Immunofluorescence Staining of Paraffin Sections

The paraffin sections of study part II were handled as described by de Buhr et al. [39]. Briefly, after dewaxing, rehydration, permeabilization, and blocking, the staining was performed for one hour with a monoclonal mouse anti-DNA/histone-1 antibody (Millipore MAB3864, 0.55 mg/mL) diluted 1:100 and rabbit anti-human myeloperoxidase antibody (Dako A0398, 3.2 mg/mL) diluted 1:300 or rabbit anti-elastase antibody (Abcam, Cambridge, UK Ab1876, 10 mg/mL) diluted 1:50, with all dilutions in blocking buffer. An isotype control staining was achieved by using IgG2α from murine myeloma (Sigma-Aldrich M5409-.1MG, 0.2 mg/mL) and IgG from rabbit serum (Sigma-Aldrich I5006, 10.4 mg/mL) in adjusted concentrations. Goat anti-mouse DyLight 488 (Thermo Scientific 35503, 1 mg/mL, 1:1000) and goat anti-rabbit Alexa 633 (Thermo Scientific A 21070, 2 mg/mL, 1:500) antibodies, both diluted in blocking buffer, were utilized for the secondary staining for one hour in the dark. The samples were then washed, embedded in ProLong^®^Gold with 4′,6-diamidino-2-phenylindole (DAPI) (Thermo Fisher P36931), and covered with a cover slip (Roth H878).

### 2.8. Immunofluorescence Microscopy and Analysis of NETs

A Leica TCS SP5 AOBS confocal inverted-base fluorescence microscope with an HCX PL APO 40× 0.75–1.25 oil immersion objective and an HCX PL APO lambda blue × 63 1.40 oil immersion UV objective was used to record the samples in study part II. The settings were adjusted with respective isotype controls. For the NET induction assays, in each sample, a minimum of six randomly selected images per independent experiment were taken and used for quantifying the activated NET-positive cells. On the basis of the inhomogeneous structure and size of nuclei in horse neutrophils, the size of nuclei was not used as a parameter for the NET-positive cells. Instead, a cell was defined as positive when the nucleus of the cell showed a positive signal for DNA/histone-1 staining as NET marker. For each individual experiment and horse, a respective negative control was analyzed in parallel.

### 2.9. ELISA for eCATH

Serum from study part I and VBF samples of horses with healthy eyes and ERU-diseased horses from study part II were examined with a Horse Cathelicidin Antimicrobial Peptide ELISA Kit (Biozol, Eching, Germany ASB-OKEH03902) in accordance with the manufacturer’s recommendations for evaluating the amount of equine cathelicidins in those samples.

### 2.10. Electron Microscopy

Fresh isolated blood-derived equine neutrophils were put in Eppendorf tubes at a concentration of 1 × 10^6^ cells per tube and centrifuged for five minutes at 400 *g* at room temperature. After the supernatant was pipetted off, 100 µL VBF of one animal with healthy eyes (study part II) or 100 µL RPMI (gibco 11835063) was added to the pellet. Then, 10 µM of equine cathelicidins (eCATH 1 or 2, see above), diluted in VBF or RPMI, was added in a 2:1 ratio. This was incubated for 240 min at 37 °C with 5% CO_2_. The subsequent steps were additionally performed with 2 mL VBF of one ERU-diseased horse from study part II. The samples were centrifuged for five minutes at 400 g at room temperature, after which the supernatant was discarded and 250 µL 2.5% (vol/vol) glutaraldehyde in 0.1 M sodium cacodylate (pH 7.2) was added to the pellet. They were then post fixed with 1% osmium tetroxide (wt/vol) and 0.15 M sodium cacodylate (pH 7.2) for 1 h at 4 °C, washed, and further processed for electron microscopy.

For transmission electron microscopy, the fixed and washed samples were subsequently dehydrated in ethanol and further processed for standard Epon embedding. Sections were cut with an LKB ultratome and mounted on Formvar-coated copper grids. The sections were post fixed with uranyl acetate and lead citrate and examined in a Philips/FEI CM100 BioTwin transmission electron microscope operated at a 60-kV accelerating voltage. Images were recorded with a Gatan Multiscan 791 charge-coupled device camera.

The ultrathin sections were stained with uranyl acetate (Laurylab, Saint Fons, France) and lead citrate (Laurylab). Immunolabeling of thin sections after antigen unmasking with sodium metaperiodate (Merck) [40] with gold-labeled anti-TNFα (BBInternational, Cardiff, UK) was performed as described previously [41], with the modification that Aurion-BSA (Aurion, Wageningen, The Netherlands) was used as a blocking agent.

### 2.11. Statistical Analysis

Data were analyzed using Excel 2010 and 2016 (Microsoft) and GraphPad Prism version 8.0.1. (GraphPad Software). Normal distribution of data was verified using the Kolmogorov-Smirnov normality test (GraphPad software, San Diego, CA, USA) prior to statistical analysis. Differences between groups were analyzed as described in the figure legends (* *p* < 0.05, ** *p* < 0.01, *** *p* < 0.001, **** *p* < 0.0001).

## 3. Results

### 3.1. More NET Markers in Serum of ERU-Diseased Horses

To test whether NETs could contribute to the pathogenesis of ERU, an initial screening for NET markers using serum from ERU-diseased horses and animals with healthy eyes was performed. As NETs consist of DNA, the amount of free DNA was evaluated by a Pico Green assay in the serum. Hereby, it needs to be taken into account that free DNA in serum might originate from sources other than neutrophils [42,43]. Thus, the results only represent first indications of the possible presence of NETs in ERU-diseased horses. The average amount of free DNA was significantly different with 4.0 ± 6.1 µg/mL in the ERU serum and 1.9 ± 2.7 µg/mL in the serum of horses with healthy eyes (Figure 1a). As mentioned in the introduction, NET formation is often associated with an increased release of nucleases by the host in order to maintain a balance between NET formation and NET elimination [26]. An analysis of nuclease activity in the same serum also revealed a significant difference between samples from horses with healthy eyes and ERU-diseased horses. In the ERU serum, a significantly higher relative nuclease activity of 102 ± 22.6 was observed, compared with 83.5 ± 31.2 in the control serum (Figure 1b). However, this finding harbored a first suspicion that NETs contribute to the pathogenesis of ERU-diseased horses.

### 3.2. NET Detection Ex Vivo

In order to confirm the presence of NETs in the eyes of ERU-diseased horses, we performed a series of immunofluorescence stainings for NET markers in VBF. The NETs were identified through a positive signal for DNA/histone-1-complexes and a colocalization of the signal with myeloperoxidase. NET structures were present ex vivo in the VBF of two of three ERU patients (Figure 2), but not in those animals with healthy eyes. All eyes were additionally examined histologically to confirm or exclude ERU (Table A2). As shown in Figure 3, several of the activated cells formed vesicles rather than NET fibers containing DNA/histone-1-complexes. This vesicular release is a special form of extracellular trap formation shown in vitro in cases of response to *Staphylococcus aureus* in human neutrophils [23], and is induced by *Listeria* in murine microglia [44].

We additionally examined VBF of one ERU-diseased horse (horse H) via transmission electron microscopy for a more detailed analysis of NET-releasing cells ex vivo. Hereby, H3-cit and elastase were labeled by immunogold staining. As shown in Figure 4, neutrophils were detected with clear signs of NETosis, for example, disruption of the nuclear membrane and release of extracellular traps. Besides cells undergoing NETosis, the formation of nuclear vesicles positive for H3-cit and elastase in neutrophils that were found inside the VBF of ERU-diseased horses was also confirmed. The magnified pictures in Figure 4e show the vesicular NET formation in several neutrophils.

The presence of NET markers and neutrophils during NETosis in VBF from ERU-diseased horses confirmed our hypothesis that NETs play a role in ERU pathogenesis.

### 3.3. More Activated Neutrophils in ERU-Diseased Horses Despite Nuclease Activity

To obtain a deeper insight into the influence of VBF on NET formation, we exposed fresh neutrophils isolated from healthy horses to VBF from animals with healthy eyes compared with ERU-diseased horses. A significant increase in the percentage of activated cells could be identified in VBF of the ERU-diseased horses compared with those with healthy eyes (Figure 5a,b). The incubation of neutrophils with methyl-β-cyclodextrin diluted in VBF, serving as positive control, also led to a significantly higher NET formation in ERU-diseased horses compared with those with healthy eyes.

As host nucleases are important for NET elimination and might also be present in the VBF, we evaluated their corresponding activities in the VBF samples by means of a DNase activity test. The numbers and letters below the gels and in the graph indicate the corresponding samples tested at the same time for NET formation. While the VBF of horses with healthy eyes were all negative, most of the VBF of ERU-diseased horses showed distinct nuclease activity (Figure 5c,d). These data are in good accordance with the detected increased nuclease values in sera from ERU-diseased horses shown in Figure 1. Although this difference between the amount of activated cells in VBF compared with VBF without nuclease activity was not significant, a tendency for higher nuclease activity in VBF of ERU-diseased horses was detectable (*p* = 0.068). However, it is important to highlight that a high amount of NET formation was found in neutrophils treated with VBF of ERU-diseased horses, despite the presence of the shown higher nuclease activity. These data again confirmed that NETs might play a role during ERU pathogenesis.

### 3.4. Influence of Equine Cathelicidins

#### 3.4.1. NET Induction with Equine Cathelicidins

After having demonstrated the presence of NETs in ERU-diseased horses, we aimed to analyze possible NET-inducing factors during ERU pathogenesis. Because cathelicidins are described as an NET-inducing factor in humans [45], and because the cathelicidin-modulating factor IL-17 is linked to ERU [11], it may be assumed that cathelicidins might be involved in NET formation in ERU-diseased horses. Therefore, we focused on further characterizing the impact of equine cathelicidins on NET release in horses (Figure 6). Fresh isolated blood derived neutrophils from donors with healthy eyes were incubated with three different eCATH at amounts of 5 µM and 10 µM. These concentrations were chosen because human cathelicidin has been shown to induce NETs in human neutrophils in such concentrations [45]. After a 240 min incubation period with RPMI and eCATH, the cathelicidins eCATH 1 and 2 induced significantly more NETs compared with a negative control with RPMI alone (Figure 6a). Moreover, a concentration of 10 µM activated more cells than 5 µM. However, eCATH 3 did not make any difference, independent of the concentration used. Similarly, as shown ex vivo in VBF (Figure 3), several of the activated cells formed vesicles rather than NET fibers (Figure 6c). The results of an analysis of equine neutrophils incubated over 120 min with eCATH 1, 10 µM diluted in RPMI via transmission electron microscopy, and neutrophils incubated only in RPMI are depicted in Figure 6d. Again, neutrophils during NETosis and with extracellular traps were detected. Furthermore, nuclear vesicles positive for H3-cit and elastase were found. With these results, a potential influence of eCATH 1 and 2 on NET formation in horses was suggested.

#### 3.4.2. More Cathelicidins in VBF of ERU-Diseased Horses

After showing the impact of eCATH 1 and 2 on horse neutrophils in vitro, we subsequently investigated their concentrations in VBF of horses with healthy eyes and ERU patients. For this purpose, we performed a horse cathelicidin antimicrobial peptide ELISA with VBF samples from animals with healthy eyes and ERU patients (Figure 6b). The values of ERU-diseased animals showed a significantly higher amount of cathelicidins. The mean values hereby were 2.8 ng/mL for the horses with healthy eyes and 3.3 ng/mL for the diseased horses. However, these concentrations were distinctly lower than that used in NET induction assays. No difference was seen for serum levels of cathelicidins (Figure A2a).

## 4. Discussion

Numerous indications of an immune-mediated pathogenesis were reported in ERU. As neutrophil infiltration was characterized during the course of this disease [46,47,48], we investigated their contribution in terms of NET production in ERU patients. Our results demonstrate that significantly more NET markers are present in serum of ERU-diseased horses compared with horses with healthy eyes (Figure 1). However, these markers are not specific for NETs and could also originate, for example, from necrotic or apoptotic cells, or be present owing to diverse pathological states [42,43]. Other serum NET markers, such as myeloperoxidase, neutrophil elastase, myeloid-related protein, or nucleosome levels, exist, but free DNA was shown to be the most promising among these serum NET markers [49]. Another possibility would be to perform a serum analysis for H3-cit, which is considered as the most specific marker [50,51]. Nevertheless, we subsequently focused on more specific NET markers together with the visualization of the cells inside the affected eye. Such NET markers as H3-cit, DNA/histone complexes, and myeloperoxidase or elastase are present in activated neutrophils or nuclear vesicles of VBF from most ERU-diseased horses (Figure 2, Figure 3 and Figure 4) and their VBF activates more neutrophils than VBF of horses with healthy eyes (Figure 5a). We did not find NET markers in VBF of every diseased horse, which might be because of the sample size or the fact that neutrophils are rather involved in the onset of the disease than in the later phases [46]. Our samples for the ex vivo NET detection were all derived from horses with several reported acute phases in their history, leading to a recurrent condition with eye damage, so the eye had to be removed for therapeutical reasons at a later stage of the disease. This chronic state of ERU, when the eye was removed, is described as being characterized by lymphoplasmatic cells [46], which was confirmed in our histopathological examinations (Table A2). Owing to this stage of the disease, we might not have found NET fibers in the three ERU-diseased horses included in the in situ analysis. However, in the Appendix A (Figure A3), we show an in situ analysis derived from a horse with acute uveitis (no ERU). This horse shows clear NET fibers inside the ciliary body, demonstrating the possibility of NET release inside this tissue during acute phases of inflammation.

In summary, our findings lead to the hypothesis that ERU-diseased horses develop more NET markers and that NETs may contribute to the pathogenesis thereof. This is in accordance with studies in which a pathological component of NETs was already found in other diseases. For instance, autoimmune antibodies against intracellular antigens bind NETs as they are exposing intracellular components to the extracellular space [52]. Such antibodies were, for example, detected in rheumatoid arthritis, in this particular case against citrullinated histone [27]. Furthermore, autoantibodies themselves can in turn induce NETs, leading to a vicious circle of NETosis and autoantibody production, maintaining the inflammation [27]. Among others, protein excess through release of neutrophil elastase and an activation of the complement system by non-degraded NET particles are described mechanisms [53]. Nevertheless, NETosis also appears as a mixed blessing with both beneficial and detrimental effects at once. This was described by Thanabalasuriar et al. [29] in a *Pseudomonas aeruginosa* ocular surface infection. By building up a dead zone underneath the biofilm, NETs limit the bacterial dissemination into the brain, but coincidently cause a more severe ocular clinic. Hence, a detrimental effect of NETs also arising in ERU is plausible.

Besides clear signs of NETosis, hints of vesicular release of NETs were also detectable in ERU-diseased horses. The vesicular NET formation was described as a vital NET formation [23]. This mechanism was identified as a response of neutrophils to the presence of *Staphylococcus aureus*. With the present study, we identified a similar picture ex vivo as well as in vitro in the VBF of an ERU-diseased horse and after eCATH stimulation in vitro (Figure 3, Figure 4 and Figure 5). Interestingly, by means of electron microscopy (EM) analysis, the different forms of NET formation were identified in the same sample. Alongside the vesicular NET formation, neutrophils with a clear NET release and a membrane disruption were identified. Therefore, this leads to the hypothesis that neutrophils are primed or programmed in different ways, eventually by different stimuli during ERU pathogenesis.

As potential NET inducing factors, we investigated the amount and effects of the equine cathelicidins 1-3 (Figure 6), because of the described genetic link of the cathelicidin-modulating factor IL-17 to ERU [11]. Antimicrobial peptides in general and cathelicidins specifically have been reported to induce NETosis and stabilize NETs [54]. A detrimental effect thereof can be seen in mice, where cathelicidin in NETs promotes atherosclerosis through activation of plasmacytoid dendritic cells [55]. Additionally, defects in processing, expression, or function were found in several human inflammatory skin diseases. Hereby, an overexpression can be just as detrimental as a deficient antimicrobial barrier function caused by downregulation [56].

In horses, three different cathelicidin sequences have been identified at the RNA level, but only two mature peptides [36]. However, we performed the assays with all three equine cathelicidins, as the likelihood of an eCATH 1 expression under specific conditions or in other cell types than myeloid cells was suggested by Bruhn et al. [57]. Our results show that eCATH 1 and 2, but not eCATH 3, have the ability to activate neutrophils to form NETs in RPMI (Figure 6a), as already shown for the human cathelicidin LL-37 in human neutrophils [45]. The varying results between the cathelicidins could be owing to their molecular structure. The coding regions of the eCATH peptides have different lengths and no significant sequence homology [58]. In particular, eCATH 3 has a low hydrophobic part, thus explaining its poor effects [36]. The equine cathelicidins have also been reported to be different regarding their antimicrobial properties. A synthetic eCATH 1 was hereby most bactericidal, followed by eCATH 2. The reaction of eCATH 3 is highly dependent on its environment, as it effectively combats pathogens in low salt concentrations, but shows an inhibited activity in physiologic salt medium [36]. Interestingly, the used low-ionic media contained a salt concentration of 100 mM NaCl, similar to the RPMI used in our experiments (103 mM). Nonetheless, we still have not discovered a NET inducing activity for eCATH 3 in RPMI. This is in line with the finding that bactericidal activity of cationic antimicrobial peptides does not always correlate with the NET-inducing or NET-stabilizing activity, as shown by Neumann et al. [45,54]. An influence of sodium chloride has likewise been reported for the interaction between DNA and LL-37 in humans. Salt promotes the dissociation of LL-37 from extracellular DNA through interference of the ionic interactions between DNA phosphate groups and amino acids of the AMP [59]. The amount of sodium chloride in equine VBF is 17.7% higher [60] than in RPMI, but still only about a third of the lowest amount tested by Lande et al. [59]. Moreover, the pH is an influencing factor on the activity of several human AMPs, including the human cathelicidin LL-37, where an acidic surrounding reduces the function [54,61]. In line with this discussion is the finding that the NET-inducing effect of eCATH 1 and 2 is lost in the presence of VBF of healthy or diseased horses (Figure A2b,c). Thus, it still remains questionable if the eCATH does in fact contribute to NET induction in ERU-diseased horses. To investigate this in detail, the effects of an inhibitor of eCATH on NET formation could be examined, but no effective antibodies are currently available.

Additionally, the NET-inducing capacity of eCATH 1 and 2 was evident at higher concentrations (5 µM and 10 µM) compared with the measured lower amounts of eCATH in the VBF. However, it has to be mentioned that these measured samples were derived from eyes in sub-clinical stages. The actual amounts inside an acutely inflamed eye are likely to be much higher, as the expression of LL-37 is induced during inflammation [62], leading, for example, to median levels up to 300 µM in psoriatic skin lesions [63]. On the basis of these results, we assume that besides cathelicidins, other factors, such as pathogens, might also contribute to NET formation during ERU.

One possible major contributor to the pathogenesis of ERU is *Leptospira*, which can be found in about 60% [12,13,14] of ERU-diseased horses and, importantly, not in VBF of horses with healthy eyes. Nothing is known about their influence in the context of NETs in ERU so far. However, as *Leptospira* have been proven to induce NETs, for example, in humans, mice, and cattle [64,65], their impact on equine neutrophils remains to be determined. According to our data, there is neither a link between the *Leptospira* titer (Table A1) and the percentage of activated cells in the NET induction assay (Figure A2b,c), nor to the nuclease activity. The horse with the lowest titer even induced the highest percentage of NETs. This corresponds with the finding that both *Leptospira* positive and negative horses develop all degrees of severity of this disease [14]. All samples were positive for the serovar Grippotyphosa, which is in accordance with data describing it as being the most widely distributed serovar in Europe [13,14,66]. Nevertheless, there could be a connection between the amount of NETs in acutely inflamed eyes and living *Leptospira* inside the inflamed VBF, as our samples were derived during clinically quiescent periods. In humans, a concentration-dependent induction of NETs was found [64].

Regardless of their initiating trigger, the involvement of NETs leads to treatment options not yet included in the current therapy approaches of ERU. In other diseases regarding NETosis, the application of DNase, heparin, or anti-VEGF was successful. DNase degrades extracellular DNA and is thus able to clear NETs [67]. An impairment of this enzyme has been shown to contribute to disease progression, such as lupus nephritis [18]. For instance, DNase treatment is established in cystic fibrosis therapy in humans [53]. Additionally, DNase eye drops had beneficial effects in dry eye disease, like significantly reduced corneal defects and mucoid debris [67,68]. Barliya et al. [31] have proven a decreasing effect of DNase ex vivo, treating NETs in cryosections of murine eyes inflamed owing to injection of IL-8 or TNF-α. Consequently, they discuss a potential therapeutic use of DNase in intraocular inflammatory processes. Another approach to degrade NETs in an ocular disease has been performed in ocular graft-versus-host disease using heparin [30]. This drug removes histones from NETs owing to its high negative charge, resulting in destabilization [30]. In a sub-anticoagulant dose, it not only reduced the clinical symptoms and led to fewer amounts of inflammatory cytokines, but also increased cell proliferation. Furthermore, a decrease in NET formation in proliferative diabetic retinopathy has been demonstrated after intraocular anti-VEGF injection [32].

Administering these drugs could consequently help to show the extent to which NETs contribute to the development of ERU and whether a DNase or heparin treatment could be used as a new therapy. Nevertheless, beforehand, further studies using in vivo experiments or complex 3D culture system, which closely mimic the cellular interactions in the host, are needed. For this purpose, a 3D cell culture system should be used to separate the blood from the VBF compartment of the eye. Thereby, the blood-retina barrier, over which the neutrophils need to migrate, would be mimicked. In humans, the retinal pigment epithelium cell line ARPE-19 has already been used on filters to simulate this barrier [69]. Nevertheless, as the presented study focused on ERU in horses, an equine system is needed to investigate this aspect in the future. Additionally, a cell culture system could prospectively enable specific knock-outs of antimicrobial peptide production via CRISPR/Cas. Furthermore, in this line, a conceivable detrimental effect of NETs on the retinal barrier could be investigated with this system. NETs could probably act detrimentally towards this epithelial barrier, as they also display a destructive impact on other epithelial tissue. For instance, NETs cause cell death in lung epithelial cells in a concentration-dependent manner [70]. As this effect was independent of DNase treatment, but dependent to varying degrees on anti-histone antibodies, polysialic acid, and MPO inhibitor, Saffarzadeh et al. [70] suggest a mediation of cytotoxicity mainly by histones and MPO instead of the DNA components of NETs. This corresponds to the finding that NETs lead to corneal and conjunctival epitheliopathy, which was treatable with heparin [30].

In conclusion, our data suggest an involvement of NETs within the pathogenesis of ERU. Meanwhile, the equine cathelicidins seem to not be the determining factor, but only a contributing one. The association of NETs with eye diseases remains an important field that has yet to be investigated. Thereby, attention should be given to molecular processes in regards to the development of future therapeutic approaches, as recommended by Estúa-Acosta et al. [71]. Possible molecular processes to focus on would be the signaling pathways and clearance of NETs during these ocular pathologies [71]. Furthermore, the discussed open questions should be answered in the future using complex 3D cell culture system that mimic cellular interactions in the host.

## Figures and Tables

**Figure 1 cells-08-01528-f001:**
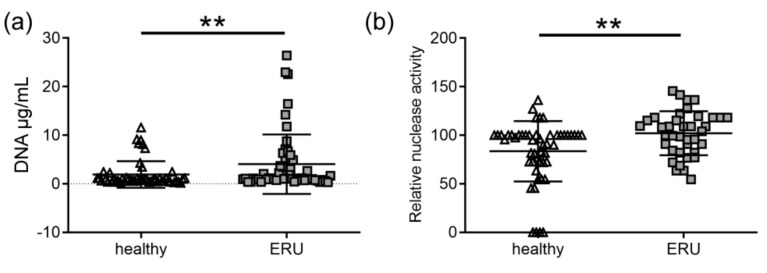
Free extracellular DNA and relative nuclease activity in serum. The amounts of free extracellular DNA (**a**) and relative nuclease activity (**b**) were analyzed in serum from 52 equine recurrent uveitis (ERU) patients and 51 eye-healthy control animals (study part I, Munich). In (**a**), a Pico Green assay was used. Significantly more free DNA was detected in the ERU serum compared with the control serum. In (**b**), the relative nuclease activity was calculated to a control digestion with DNase I. The individual values and the mean value of the respective groups are given. In the ERU serum, a significantly higher nuclease activity was observed compared with the control serum. In (**a**,**b**) the individual values and the mean value ± SD of the respective groups are shown (** *p* < 0.01, two-tailed Mann–Whitney test).

**Figure 2 cells-08-01528-f002:**
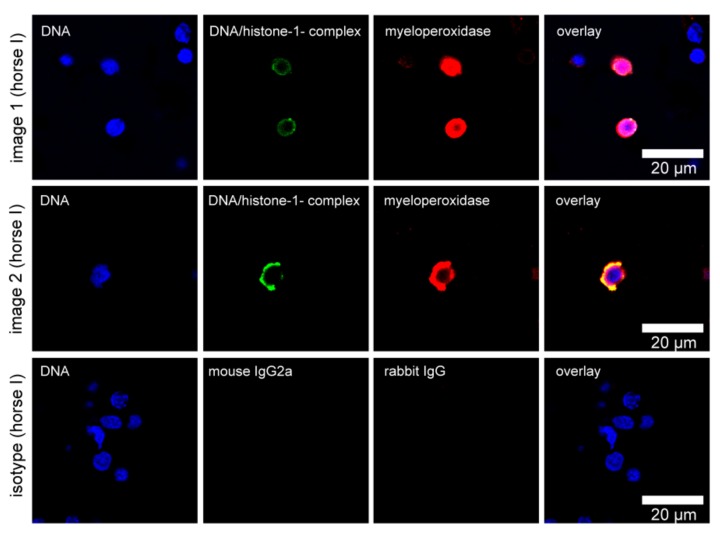
Ex vivo neutrophil extracellular trap (NET) detection in vitreous body fluid (VBF) of ERU-diseased horses (study part II, Hannover). The VBF of enucleated eyes from ERU-diseased horses was analyzed after cytospin and immunofluorescence staining with a confocal microscope. NET staining for immunofluorescence microscopy of the cells contained in the VBF was conducted (blue = DNA (Hoechst), green = DNA/histone-1-complexes, and red = myeloperoxidase). Representative images of cells from one animal (horse I) are shown in rows 1 and 2. Cells in image one are stained in the center of the cells, whereas in image 2, the staining of DNA/histone-1-complexes and myeloperoxidase surrounds the cell. The respective isotype control is presented in row 3.

**Figure 3 cells-08-01528-f003:**
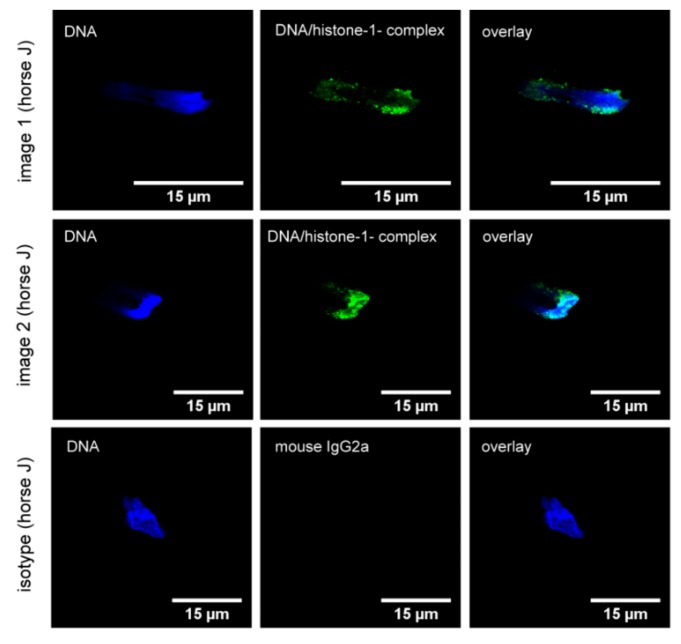
Ex vivo NET detection in VBF of ERU-diseased horses (study part II, Hannover). The VBF of enucleated eyes from ERU-diseased horses was analyzed for cells and NET markers after cytospin and immunofluorescence staining with a confocal microscope (blue = DNA, green = DNA/histone-1-complexes). Images of different cells from one animal (horse J) are shown in images 1 and 2, where the activated cells formed vesicles rather than NET fibers. The respective isotype is presented in row 3.

**Figure 4 cells-08-01528-f004:**
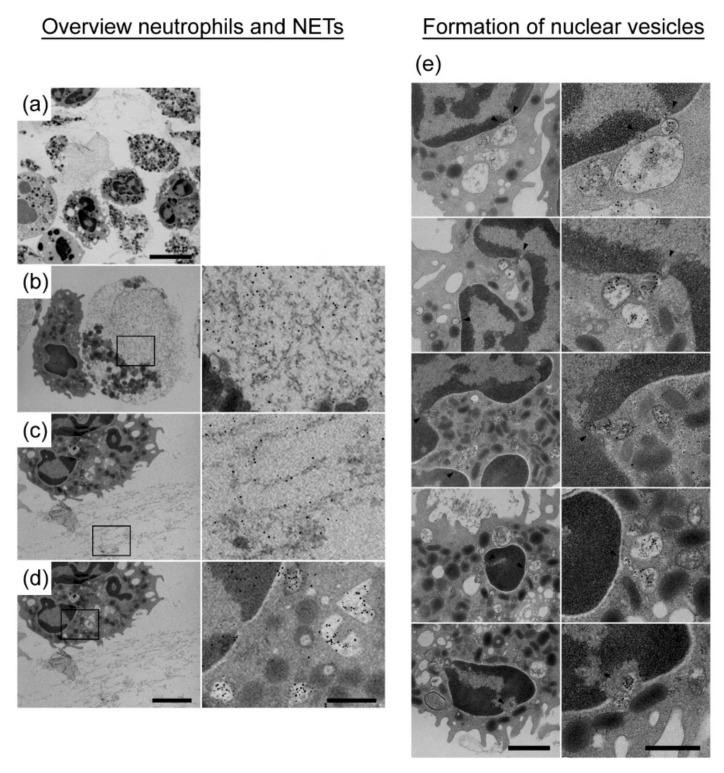
Ex vivo NET detection via transmission electron microscopy in VBF of an ERU-diseased horse (study part II, Hannover). The VBF of an enucleated eye from an ERU-diseased horse (horse H) was analyzed with a transmission electron microscope. (**a**) overview, (**b**) neutrophil during NETosis (**c**) neutrophil with an extracellular trap, and (**d**) neutrophil with nuclear vesicles. In (**a**–**d**), the right column displays respective magnifications with immunogold labeling (5 nm gold/H3-cit, 10 nm gold/elastase). (**e**) All images show the formation of H3-cit and elastase positive nuclear vesicles (5 nm gold/H3-cit, 10 nm gold/elastase). Arrows mark the formation of nuclear vesicles. On the right-hand side, the right column displays respective magnifications. Scale bar size: (**a**) = 5 µm, (**b**–**d**) left-hand side = 2 µm, (**b**,**d**) right-hand side = 200 nm, (**e**) left-hand side = 1 µm, and (**e**) right-hand side = 500 nm.

**Figure 5 cells-08-01528-f005:**
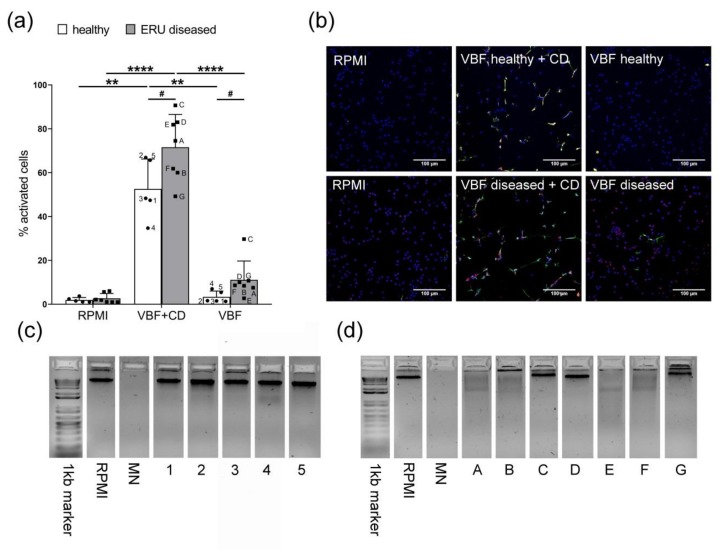
Activation of neutrophils by VBF of horses with healthy eyes and ERU-diseased horses (VBF diseased) and respective nuclease activities (study part II, Hannover). (**a**) After a 240 min incubation period of fresh isolated blood derived neutrophils with VBF, significantly more activated neutrophils were detected in samples from ERU-diseased horses. Roswell Park Memorial Institute medium (RPMI) was used as negative control, with methyl-β-cyclodextrin (CD) diluted in VBF as positive control. The bars represent mean ± SD of five (healthy eyes) or seven (ERU-diseased) independent experiments. Statistical analysis: # represents results of a one-tailed unpaired student’s *t*-test; * represents results of one-way analysis of variance (ANOVA), followed by Tukey’s multiple comparison test. *p*-values of # *p* < 0.05, ** *p* < 0.005, and **** *p* < 0.0001 were considered significant. (**b**) Representative images (horses 3 and E) of immunofluorescence analysis are presented (blue = Hoechst, green = DNA/histone-1-complexes, red = myeloperoxidase). The respective isotype control stainings and single channel pictures are shown in Figure A1. (**c**,**d**) The corresponding nuclease activities in the VBF samples were analyzed by a qualitative DNase activity test. The results of horses with healthy eyes are shown in (**c**) and of ERU-diseased horses in (**d**). RPMI was used as a negative control and micrococcal nuclease (MN) as a positive control.

**Figure 6 cells-08-01528-f006:**
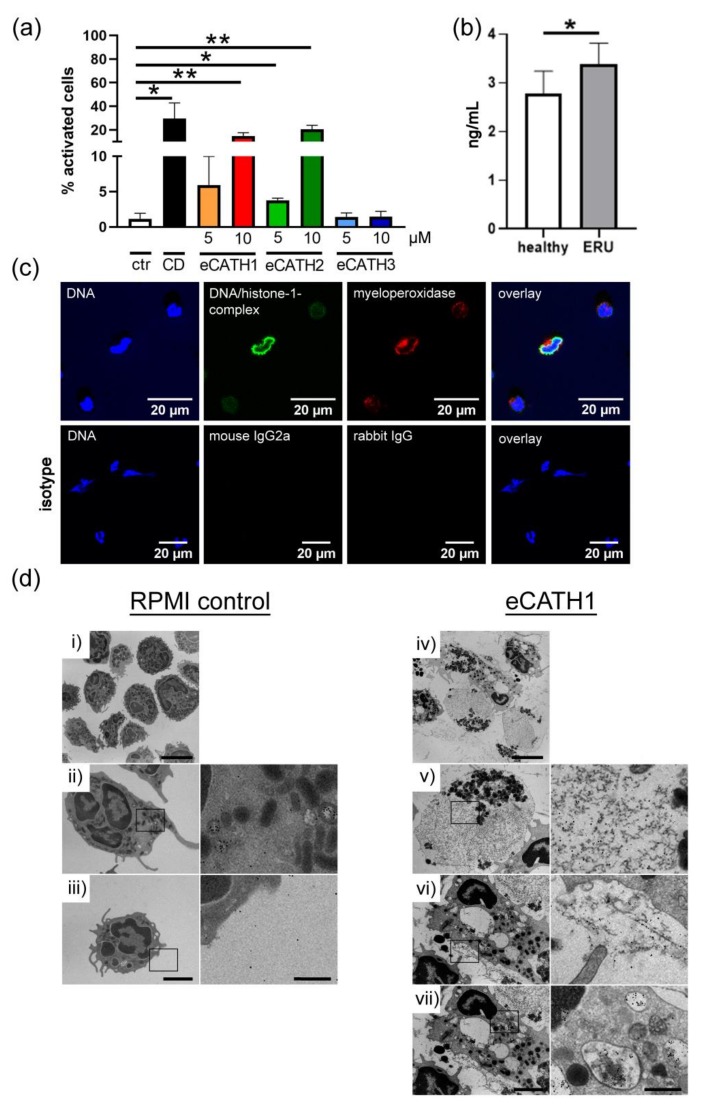
Influence of equine cathelicidins (eCATH) 1-3. (**a**) After a 240 min incubation period, eCATH 1 and 2 activated neutrophils in RPMI depending on the quantity used, whereas eCATH 3 did not lead to activation. The bars represent mean ± SD of three independent experiments. A one-tailed paired *t*-test was conducted. *p* values of * *p* < 0.05 and ** *p* < 0.01 compared with control were considered significant. (**b**) Equine cathelicidins in VBF. The amounts of equine cathelicidins in VBF of ERU-diseased (*n* = 7) and controls with healthy eyes (*n* = 5) were analyzed using a commercial ELISA. A significant increase in equine cathelicidins in VBF of ERU-diseased horses was observed. The bars represent mean ± SD. A one-tailed unpaired *t*-test was performed. *p* values of * *p* < 0.05 compared with control were considered significant. In (**c**), representative immunofluorescence images of equine neutrophils stimulated with eCATH are presented, in this case with eCATH 2 and 10 µM in RPMI (blue = Hoechst, green = DNA/histone-1-complexes, red = myeloperoxidase). The respective isotype control stainings are included. In (**d**), transmission electron microscopy images of equine neutrophils are shown. Control images from neutrophils incubated for 120 min in RPMI are presented on the left-hand side. Neutrophils stimulated with eCATH 1 (10 µM in RPMI) for 120 min are presented on the right-hand side. An overview is given in (i) and (iv). The right column in each group displays respective magnifications with immunogold labeling (5 nm gold/H3-cit, 10 nm gold/elastase). (ii) shows an intact neutrophil with elastase positive granular. (iii) shows the intact outer membrane of an unstimulated neutrophil. (v) presents a neutrophil with a high amount of H3-cit and elastase positive content in the cytosol. (vi) shows the release of an extracellular trap that is H3-cit and elastase positive. (vii) depicts a neutrophil with H3-cit and elastase positive nuclear vesicles. Scale bar size: (i) and (iv) = 5 µm, (all others) left-hand side = 2 µm, (all others) right-hand side = 200 nm.

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
