# Peer review of "Neutrophil Extracellular Traps in the Pathogenesis of Equine Recurrent Uveitis (ERU)"

_cells, 2019, doi:10.3390/cells8121528_

Round 1

Reviewer 1 Report

Manuscript number cells-639484

The paper is well written, and the aim is focused on a very important and relevant disease affecting horses. However, they are some points that need to be revised; in particular:

1) The study is organized in two main sample studies; it is very complicated to follow the manuscript with this definition of the study: what means sample 1 and sample 2? This are two separate and distinct studies, why not study 1 and study 2?

2) the Ethical approval is reported only for sample 1; in addition, it seems that the sample 1 are collected in Munich and the sample 2 in Hannover. No information’s about the sample collection period for the sample 1, while from sample 2 the collection period is reported. Similarly, no information’s are provided about ethical approval for the samples of healthy horses

3) In the first part of results the sentence “As NETs consists of DNA, the amount of free DNA was evaluated by a Pico Green assay in the serum” This sentence is very naïve, NET in blood can results from different sources as for example aging cells, or from different diseases, including cancer cells (Butt AN, Swaminathan R (August 2008). "Overview of circulating nucleic acids in plasma/serum". Annals of the New York Academy of Sciences1137 (1): 236–42); so far considering the approach to measure cDNA as direct relation with NET is not correct

Similar consideration for nuclease activity that is not specifically related to NET

4) Please better explain figure 2: “Figure 2. Ex vivo NET detection in VBF of ERU-diseased horses. The VBF of enucleated eyes from ERU-diseased horses was analyzed after cytospin and immunofluorescence staining with a confocal microscope (blue = DNA, green = DNA/histone-1-complexes, red = myeloperoxidase). Representative pictures from one animal (horse I) are shown, as well as the respective isotype (line three).

What is represented in line 2? Line 2 is quite different from line 1, what is the difference between the two lines?

5) the same consideration for figure 3, please explain

6) the experimental design of experiments related to figure 4 are not clear, please better explain the sample collection and from which kind of animals: from sample 1 or 2?

7) Figure 6, panel A is not clear, please use different color or symbols to distinguish the different experimental conditions

8) As detailed in point 1, the second sentence (related to figure 1) of discussion needs to be reformulated.

Reviewer 2 Report

Reviewer’s comment

The authors showed that neutrophil extracellular traps (NETs) formation was significantly increased in both vitreous body fluids (VBF) and serum from Equine recurrent uveitis (ERU)-diseased horses compared with those from eye-healthy horses. They also showed that in vitro NETs formation was enhanced in neutrophils incubated with VBF from ERU-diseased horses compared with that from eye-healthy horses. Furthermore, they showed that VBF concentration of cathelicidins, a possible NETs producing factor, was significantly higher in ERU-diseased horses than that in eye-healthy horses. These findings suggest that NETs formation play an important role in the pathogenesis of ERU in horses as well as human ERU. Overall the manuscript is well written. However, the following concerns need to be addressed.

Comments.

The authors showed the strong association between NETs formation and ERU; however, they did not show the causality of NETs formation in the pathogenesis of ERU. This is the weakest point in this paper. Because VBF is exchangeable in a minimally invasive way, it would be a best way to investigate the effect of DNase treatment on the progression of ERU. Considering the difficulties of in vivo experiment, it would be an alternative way to investigate the effect of NETs formation on the development of ERU using cell culture system. The authors showed that cathelicidin could induce in vitro NETs formation and that VBF concentration of cathelicidin was significantly higher in ERU-diseased horses. To further clarify the substantial role of cathelicidin on NETs formation in ERU-diseased horses, the authors should examine the effect of cathelicidin blocking peptide on the NETs formation induced by VBF of ERU-diseased horses.

Round 2

Reviewer 1 Report

the manuscript was extensively revised, however the quality of data and conclusion presentation remain low

Reviewer 2 Report

N/A